# Conformational motions and ligand-binding underlying gating and regulation in IP$_3$R channel

Guizhen Fan[1,5], Mariah R. Baker [1,5], Lara E. Terry [2], Vikas Arige[2], Muyuan Chen[3,4], Alexander B. Seryshev[1], Matthew L. Baker[1], Steven J. Ludtke [3], David I. Yule [2] ✉ & Irina I. Serysheva [1] ✉

Inositol-1,4,5-trisphosphate receptors (IP$_3$Rs) are activated by IP$_3$ and Ca$^{2+}$ and their gating is regulated by various intracellular messengers that finely tune the channel activity. Here, using single particle cryo-EM analysis we determined 3D structures of the nanodisc-reconstituted IP$_3$R1 channel in two ligand-bound states. These structures provide unprecedented details governing binding of IP$_3$, Ca$^{2+}$ and ATP, revealing conformational changes that couple ligand-binding to channel opening. Using a deep-learning approach and 3D variability analysis we extracted molecular motions of the key protein domains from cryo-EM density data. We find that IP$_3$ binding relies upon intrinsic flexibility of the ARM2 domain in the tetrameric channel. Our results highlight a key role of dynamic side chains in regulating gating behavior of IP$_3$R channels. This work represents a stepping-stone to developing mechanistic understanding of conformational pathways underlying ligand-binding, activation and regulation of the channel.

Inositol-1,4,5-trisphosphate receptors (IP$_3$Rs) are ubiquitous Ca$^{2+}$ channels predominantly localized to the endoplasmic reticulum (ER) membranes. IP$_3$Rs play fundamental roles in the physiology of a diverse array of organisms as they mediate the release of Ca$^{2+}$ from ER Ca$^{2+}$ stores in response to diverse physiological stimuli. Decades of studies have revealed the central role that IP$_3$Rs have in generating complex intracellular Ca$^{2+}$ signals, which are responsible for many complex physiological processes ranging from gene transcription and secretion to the more enigmatic brain activities such as learning and memory. Dysfunction of IP$_3$Rs leads to aberrant Ca$^{2+}$ signaling that is associated with a multitude of human diseases such as Alzheimer's disease, hereditary ataxias, cardiac hypertrophy, heart failure, Parkinson's and Huntington's diseases, atherosclerosis, hypertension and some migraines[1–3].

In mammals, there are three subtypes of IP$_3$R defined by characteristic features encoded by separate genes, and among them the type 1 IP$_3$R (IP$_3$R1) is predominantly expressed in the central nervous system, especially in Purkinje cells of the cerebellum. All IP$_3$R channels are tetrameric assemblies of monomers of either identical or different subtypes, resulting in channel complexes with a wide range of functional properties. Each IP$_3$R subunit comprises the pore-forming transmembrane domains (TMDs) and cytoplasmic (CY) domains containing binding sites for ligands and multiple modulators of the channel. These nonselective Ca$^{2+}$ channels are tightly linked to essential phospholipase C-signaling pathways that mediate generation of IP$_3$ and control a wide range of Ca$^{2+}$-dependent cellular processes in a highly coordinated manner. Complex cross talk occurring between these pathways and IP$_3$R channels leads to precise regulation of intracellular Ca$^{2+}$ levels. However, an in-depth understanding of the

[1]Department of Biochemistry and Molecular Biology, Structural Biology Imaging Center, McGovern Medical School at The University of Texas Health Science Center at Houston, 6431Fannin Street, Houston, TX, USA. [2]Department of Pharmacology and Physiology, School of Medicine and Dentistry, University of Rochester, Rochester, NY, USA. [3]Verna and Marrs McLean Department of Biochemistry and Molecular Biology, Baylor College of Medicine, One Baylor Plaza, Houston, TX, USA. [4]Present address: SLAC National Accelerator Laboratory, Menlo Park, CA, USA. [5]These authors contributed equally: Guizhen Fan, Mariah R. Baker. ✉e-mail: David_Yule@URMC.Rochester.edu; irina.i.serysheva@uth.tmc.edu

comprehensive mechanism governing control of intracellular $Ca^{2+}$ dynamics is still lacking.

IP₃Rs work as signaling hubs through which diverse cellular inputs like IP₃, $Ca^{2+}$, ATP, thiol modifications, phosphorylation and interacting proteins are processed and then integrated to result in cytosolic $Ca^{2+}$ signals with precise temporal and spatial characteristics. IP₃ and $Ca^{2+}$ are the primary agonists of IP₃R channels and both are required for channel opening. Moreover, to fulfill their many physiological roles in vivo, IP₃Rs associate with an array of regulatory molecules ranging from ions and small chemical compounds to proteins[4]. Unique interactions with these intracellular messengers contribute to the specificity, duration and shape of $Ca^{2+}$ signals generated by IP₃Rs and the channel's capacity to integrate signals from different pathways. A single IP₃ binding site exists at the N-terminus on each IP₃R subunit and functional studies suggest that binding of IP₃ to each monomer is required for channel opening[5]. Important, IP₃ has little effect in the absence of its co-agonist $Ca^{2+}$. Multiple $Ca^{2+}$ binding sites have also been predicted[6,7]. IP₃R channel opening is modulated with a biphasic dependence on cytosolic $Ca^{2+}$ concentrations, suggesting that the channel has distinct types of $Ca^{2+}$ binding sites for activation and inhibition. Electrophysiological studies showed that the IP₃-evoked activity of IP₃R is enhanced by modest increases in cytosolic $Ca^{2+}$ concentration in the sub-µM range (<1 µM), while higher micromolar concentrations of $Ca^{2+}$ are inhibitory to IP₃ responses[8,9]. ATP is known to enhance the IP₃-induced $Ca^{2+}$ release, by specific binding to IP₃R[10–15].

Despite more than three decades since the discovery of IP₃Rs, the detailed molecular and structural mechanisms underlying the complex interplay between ligands, allosteric modulators and channel gating remains unresolved. With the resolution revolution in single particle cryogenic electron microscopy (cryo-EM), there have been remarkable advances in structural characterization of IP₃Rs[16–21]. These studies reported that the IP₃R structure undergoes conformational changes upon ligand binding, suggesting structural flexibility that allows the channel to switch from a closed state, capable of interacting with its ligands such as IP₃ and $Ca^{2+}$, to an open state, capable of transferring $Ca^{2+}$ ions across the ER membrane. However, all of these 3D cryo-EM structures represent defined static conformations of the channel, and the mechanistic insights are derived based on interpolations between discrete structures, each of them likely a mixture of states from a dynamic conformational ensemble. Therefore, mechanistically it remains poorly understood how ligands engage the ligand-binding domains and how multiple signals are coordinated and processed within the IP₃R channel. This implies the necessity to consider the dynamic conformational landscape of the channel protein in determining the molecular mechanisms underlying its function.

In this work, we determined the structures of the IP₃R channel trapped in physiologically relevant states using key ligands that target channel gating. Here, we report cryo-EM structures of IP₃R1 bound to IP₃, ATP, $Ca^{2+}$ and IP₃R1 bound to $Ca^{2+}$ alone at overall resolutions of 3.50 Å and 3.26 Å, respectively. In addition, we used a deep neural network approach[22] and 3D variability analysis[23] to extract functionally relevant conformational motions of the IP₃R1 domains directly from 2D cryo-EM images. The ratcheting mechanism for IP₃-binding was built via deep-learning approach and corroborated by experimental single-particle cryo-EM analysis. This combined approach allowed for a mechanistic understanding of how conformational motions of the ARM2 domain are coupled with structural changes in the IP₃-binding pocket in the context of the tetrameric IP₃R channel. From these studies, we correlate protein structural changes that connect the ligand-binding regulatory sites to the channel gate via the IP₃R conformational landscape. Our structural findings and hypothesis were validated through mutagenesis and electrophysiology (see also[24]). Our study provides a structural framework for understanding the allosteric mechanisms underlying ligand-mediated IP₃R activation and regulation.

## Results

### Inherently dynamic architecture of IP₃R1

To elucidate allosteric coupling mechanisms underlying ligand-binding and gating kinetics of the IP₃R channel, we isolated native, $Ca^{2+}$ permeable IP₃R1 channels and used cryo-EM single particle analysis to solve the channel structures in distinct ligand-bound conformations. As the membrane environment is relevant for the functionality of channels, we studied IP₃R1 reconstituted into lipid nanodisc[20]. IP₃ and $Ca^{2+}$ are imperative ligands to trigger the channel opening and ATP increases the open probability of IP₃R1 by synergizing with the activating effect of these two primarily ligands[10–15]. Purified IP₃R1 binds IP₃ in a stoichiometric manner (one IP₃ per IP₃R1 monomer) with an affinity of 2–100 nM while functional IP₃ affinities vary from the nanomolar to low micromolar range with mean values for maximum effect around 10 µM, dependent on the technique used[5,25,26]. $Ca^{2+}$ regulates IP₃R1 gating with positive and negative feedback acting in a bell-shaped manner with maximum channel open probability at low µM free $Ca^{2+}$, while high µM $Ca^{2+}$ inhibits the channel[8,27,28].

Hence, the cryo-EM 3D reconstructions of the nanodisc-reconstituted IP₃R1 channels were determined in the presence of saturating IP₃ concentration (10 µM), -2 µM $Ca^{2+}$ and 1 mM ATP ($\underline{C}a^{2+}$/ $\underline{IP_3}/\underline{A}TP$-bound IP₃R1; CIA-IP₃R1), and using 20 µM $Ca^{2+}$, as the sole ligand ($Ca^{2+}$-bound IP₃R1, Ca-IP₃R1), at estimated overall resolutions 3.50 and 3.26 Å, respectively (Table 1; Supplementary Figs. 1–4, 6). The final reconstructions are in excellent agreement with the previously solved cryo-EM structures of IP₃R1[16,17,20] and of sufficient quality to allow constructing the atomic structural models for almost the entire protein enabling reliable assignment of most of the side chains (Supplementary Figs. 1–4). Cross-validation between the atomic models and the final density maps suggested that the average resolution of structures was around 3.50 Å and 3.80 Å for Ca-IP₃R1 and CIA-IP₃R1, respectively (Supplementary Figs. 2, 4).

The IP₃R structure has a large solenoid CY assembly built around the central helical bundle made of the C-terminal domains from four IP₃R1 subunits (Supplementary Fig. 6a, b). The solenoid scaffold includes domains responsible for binding of ligands and regulatory proteins and is connected via an allosteric nexus at the cytosolic-membrane interface to the TM channel assembly. Six TM helices from each subunit form the central ion-conduction pore and are well resolved in both the $Ca^{2+}$- and CIA-bound structures owing to the stabilizing effect of lipid nanodisc (Supplementary Fig. 7a, b).

Local resolution analysis showed that the central TM helices and ligand-binding domains (LBDs) had the highest resolution at 2.4-3 Å, while the cytoplasmic armadillo domain 2 (ARM2; L1103-G1567) had the lowest resolution, suggesting that this domain possesses structural flexibility (Supplementary Figs. 2c, 4c). Of note, this domain was similarly poorly resolved in all previous cryo-EM studies of IP₃Rs[16–21]. To improve map quality and local resolution in the ARM2 domain, we performed iterative focused refinements using a mask encompassing the ARM2 domain (Supplementary Figs. 1–4). The resulting cryo-EM maps are substantially improved, enabling atomic modeling of the ARM2 domain. Two distinct ARM2 conformations reflective of specific ligand-bound states of IP₃R1 were determined (Fig. 1; Supplementary Figs. 1, 3, 5). One conformation, hereafter called 'extended', was observed in the apo- (ligand free) and Ca-IP₃R1 structures. By contrast, in the CIA-IP₃R1 structure, the ARM2 domain is moved -30 Å towards the helical armadillo domain 1 (ARM1) from the same subunit (Fig. 1e) and exhibits a conformation distinct from that observed in the Ca-IP₃R1 or apo-IP₃R1, which we have termed 'retracted'.

To gain insights into the conformational landscape of IP₃R1 underlying its discrete ligand-bound states, we made use of a deep-learning based gaussian mixture model (GMM)[22]. When taken together, the set of raw 2D particles, each in a known orientation, does not just define a single structure, but a complete set of related structures embodying the motion and composition of the macromolecule in

**Table 1 | Cryo-EM data collection and refinement statistics**

| | Ca-IP₃R1 (EMD-27982) (8EAQ) | CIA-IP3R1 (EMD-27983) (8EAR) |
|---|---|---|
| **Data collection and processing** | | |
| Magnification | 130k | 130k |
| Voltage (kV) | 300 | 300 |
| Exposure time (s) | 7 | 7 |
| Electron exposure (e⁻/Å²) | 49 | 49 |
| Dose fractionation | 35 | 35 |
| Defocus range (μm) | −0.8 to −2.5 | −0.8 to −2.5 |
| Pixel size (Å) | 1.07 | 1.07 |
| Symmetry imposed | C4 | C4 |
| Initial particle images (no.) | 1,955,320 | 1,452,797 |
| Final particle images (no.) | 346,731 | 133,740 |
| Map resolution (Å) | 3.26 | 3.50 |
| FSC threshold | 0.143 | 0.143 |
| Map resolution range (Å) | 2.4-6 | 2.4-6 |
| **Refinement** | | |
| Initial model used (PDB code) | 7LHE | 7LHE |
| Model resolution (Å) FSC = 0.5 | 3.5 | 3.8 |
| FSC threshold | 0.5 | 0.5 |
| Map sharpening B factor (Å²) | −129 | −157 |
| Model composition | | |
| Non-hydrogen atoms | 77960 | 78516 |
| Protein residues | 9464 | 9552 |
| Ligands | | |
| IP₃ | - | 4 |
| ATP | - | 4 |
| Ca²⁺ | 20 | 16 |
| Zn²⁺ | 4 | 4 |
| lipids | 28 | 28 |
| B factors (Å²) | | |
| Protein | 104.42 | 155.44 |
| Ligand | 72.32 | 123.40 |
| R.m.s. deviations | | |
| Bond lengths (Å) | 0.003 | 0.003 |
| Bond angles (°) | 0.627 | 0.602 |
| Validation | | |
| MolProbity score | 1.79 | 1.84 |
| Clashscore | 9.44 | 10.74 |
| Ramachandran plot | | |
| Favored (%) | 95.86 | 95.78 |
| Allowed (%) | 3.97 | 4.05 |
| Outliers (%) | 0.17 | 0.17 |

solution. Using a set of Gaussian spheres in space as a proxy, deep-learning makes it possible to relate these 2D particles most similar to each of these 3D states. We can then reconstruct 3D maps using particle subsets, representing actual points along motion pathways, directly from the particle data without any structural interpolations. That is, each structure along a motion pathway was reconstructed purely from experimental data and the computational model is used only to select which data to include.

As expected, the IP₃R1 protein exhibits substantial dynamic motions involving both the cytoplasmic and pore domains (Supplementary Movie 1). In particular, our analysis captured that the ARM2

domain undergoes concerted motions in both ligand-bound states of IP₃R1 characterized in the current study, as well as in apo-IP₃R1 structure from our previous study[20]. It is notable that ARM2 exhibits similar motion trajectories in all three states, but with increased motion amplitude in the presence of IP₃ (Supplementary Movie 2). This observation suggests that the reversible IP₃-binding process in the IP₃R tetrameric channel is coupled to conformational motions of the ARM2 domain in some way. Furthermore, the helical domain (HD; E693-L1102) exhibits pronounced apical motions, and the linker domain (LNK; T2610-A2680) and intervening lateral domain (ILD; T2193-W2276) that form a nexus at the cytosolic-membrane interface, dilate. Noteworthy, these conformational changes are propagated to the HD-ARM3 interaction interface also undergoing structural rearrangements. In the TMD region the global movements are quite notable exhibiting a rotation of the TM assembly relative to the cytosolic LNK/ILD nexus (Supplementary Movie 3). Based on these results, we envision that the ARM2 domain undergoes structural transitions within the dynamic conformational ensemble and this domain flexibility affects the IP₃-binding pocket properties (discussed below). Overall, the observed conformational motions are in good agreement with our earlier proposed model for long-range allosteric conformational coupling between ligand-binding and activation of the channel gate[16,17].

### Conformational dynamics of ARM2 govern IP₃ binding

The IP₃-binding pocket is formed at a cleft between the β-trefoil domain, βTF2 (W226-V435) and the ARM1 domain (Fig. 2). The preceding βTF1 domain (M1-K225) works as an IP₃ binding suppressor[16,29–32]. These three domains (also known as ligand-binding domains, LBDs) form a triangular architecture at the apex of the CY assembly[17]. In the CIA-IP₃R1 structure, we observe a strong density bridging the βTF2 and ARM1 domains at the site where IP₃ is expected to bind (Fig. 2). This density accommodates the IP₃ molecule well. The P1 and P5 phosphate groups of IP₃ are predominantly coordinated by residues from the ARM1 domain and the P4 phosphate group interacts with the βTF2 residues. The CIA structure reveals that P4 forms hydrogen bonds with T267, R269, S274. P5 is accommodated within the binding pocket by residues: K508, R511, Y567, K569. In contrast, P1 points outside the IP₃-binding cleft and interacts with only R504 and R568, which is consistent with earlier studies demonstrating that this group is significant for IP₃ binding[33]. The three hydroxyl groups, O2, O3 and O6 can form hydrogen bonds with the ARM1 residues, but have a secondary role in binding of IP₃[34]. The overall geometry and composition of the IP₃-binding pocket are consistent with observations in crystallographic studies of expressed LBDs and cryo-EM studies of IP₃Rs (Supplementary Fig. 8) and have been validated by extensive site-directed mutagenesis[16–18,29–32,35].

Superimposition of a single subunit from the apo-, Ca²⁺- and CIA-bound structures shows that IP₃-binding causes rigid-body movements of all three LBDs resulting in a 4 Å closure of the cleft between the ARM1 and βTF2 domains. Notably, there are small intra-domain rearrangements in both βTF domains upon IP₃- and/or Ca²⁺ binding (1.2 Å Cα RMSD) (Fig. 2b; Supplementary Fig. 6). In contrast, the α-helices in the ARM1 domain undergo substantial shifts in the CIA structure with the entire domain moving towards the IP₃ binding pocket (Fig. 2b, c). In particular, the first α-helix in ARM1 (P437-A457) shifts 12° from its position in the Ca-IP₃R1 structure, and its C-terminal segment (G458-K462) becomes unraveled. Additionally, the two helices in the ARM1 domain (h3: R504-E512; h7: R568-K576), which residues directly contribute to the coordination of IP₃, move ~20° when engaged with IP₃.

Inspection of the interfaces between the LBDs and the ARM2 domain shows that ARM2 interacts with βTF1 and ARM1 domains from the neighboring subunit in Ca-IP₃R1, CIA-IP₃R1 and apo-IP₃R1 structures (Fig. 1, Supplementary Fig. 6). Specifically, in both

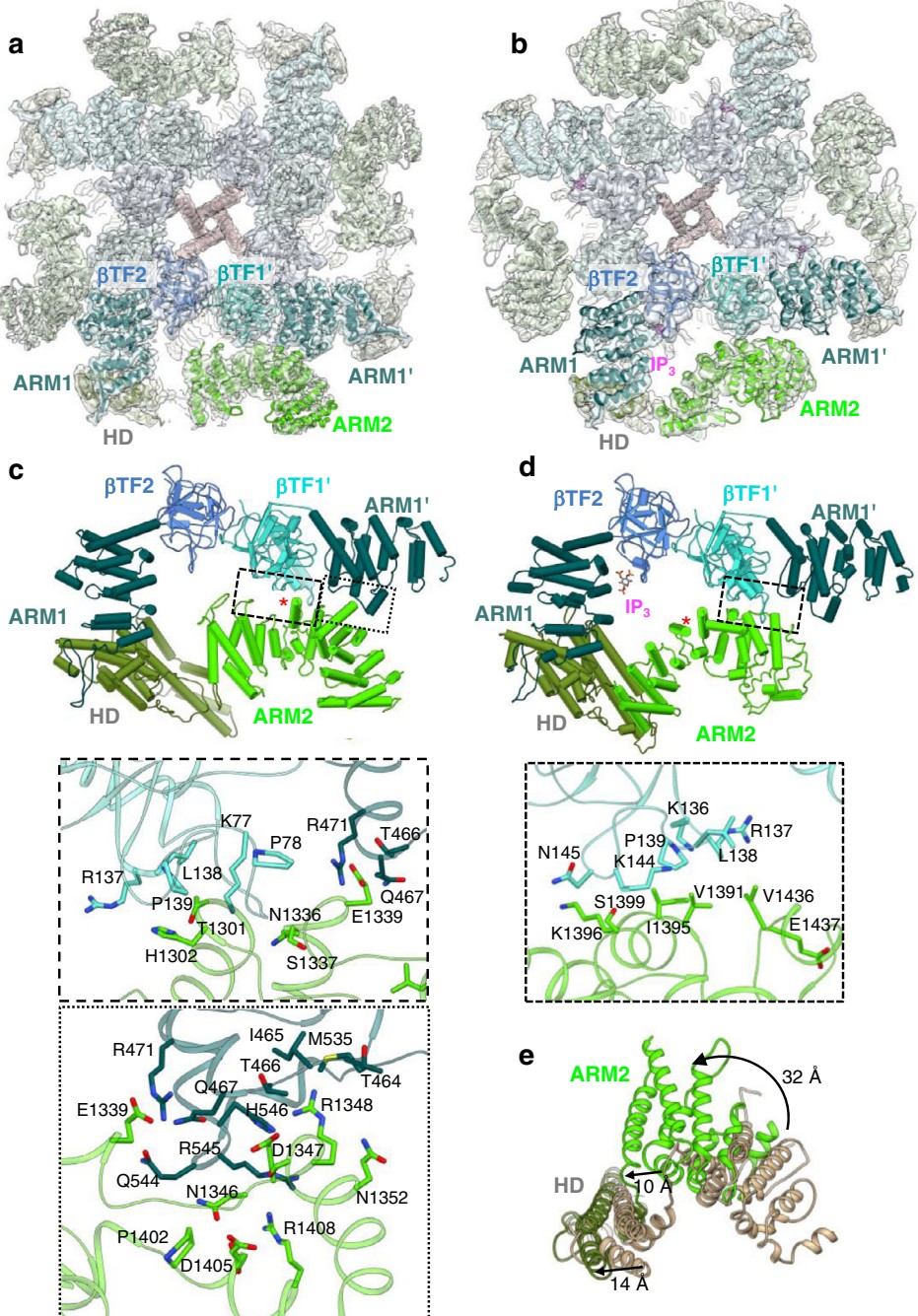

**Fig. 1 | Interactions between the ARM2 and LBDs in ligand-bound IP$_3$R1 structures. a, b** Cryo-EM density maps and corresponding models of Ca$^{2+}$ bound IP$_3$R1 (**a**) and Ca$^{2+}$/IP$_3$/ATP bound IP$_3$R1 (**b**) are viewed along central four-fold axis from the cytosol. Domains involved in inter-subunit interactions are color coded and labeled. Bound IP$_3$ density is colored magenta. The neighboring subunit is indicated by a single quote symbol. **c, d** Molecular models for ligand-binding and interfacial domains are rendered as pipes and planks for Ca-IP$_3$R1 (**c**) and CIA-IP$_3$R1 (**d**). ARM2 inter-subunit interfaces are indicated by boxes and enlarged in the lower panels. Zoomed in boxed areas depict the amino acids present at the molecular interfaces. The red asterisk marks the same helix in each structure. **e** HD and ARM2 domains from an aligned subunit from Ca-IP$_3$R1 (tan) and CIA-IP$_3$R1 (colored by domain) are superimposed. Arrows indicate 32 Å rotation of ARM2 and a 10 Å translation of HD.

apo- and Ca-IP$_3$R1 structures the βTF1-ARM2 interface is stabilized by interactions between A75-D85 and R137-A146 from βTF1 with N1336-S1337 and T1301-H1302 from ARM2 (Fig. 1c, Supplementary Fig. 6d). Additionally, an interface between the ARM1 and ARM2 domains is formed at M535-A547 and G463-R471 and S1399-R1408, N1346-Q1352 and N1336-E1339, respectively. The ARM1 residues R545, M535, R471 are in position to form hydrogen bond with residues N1346, R1348, E1399 from the ARM2 domain. We found that these specific inter-subunit interfaces are perturbed in the CIA-IP$_3$R1 structure, resulting in

the formation of a smaller interaction area at a new interface located at residues V1391-S1399 and D1433-M1438 of ARM2 and K136-N145 of βTF1 (Fig. 1d).

The functional implication of these intersubunit domain-domain interactions is, perhaps, to relieve structural constraints imposed at ARM2/βTF1 and ARM2/ARM1 interfaces that would prevent the LBDs to undergo motions crucial for rendering the IP$_3$-binding pocket conformations either suitable or unsuitable for capturing IP$_3$. As a result, the ARM1 α-helices shaping the binding pocket undergo structural

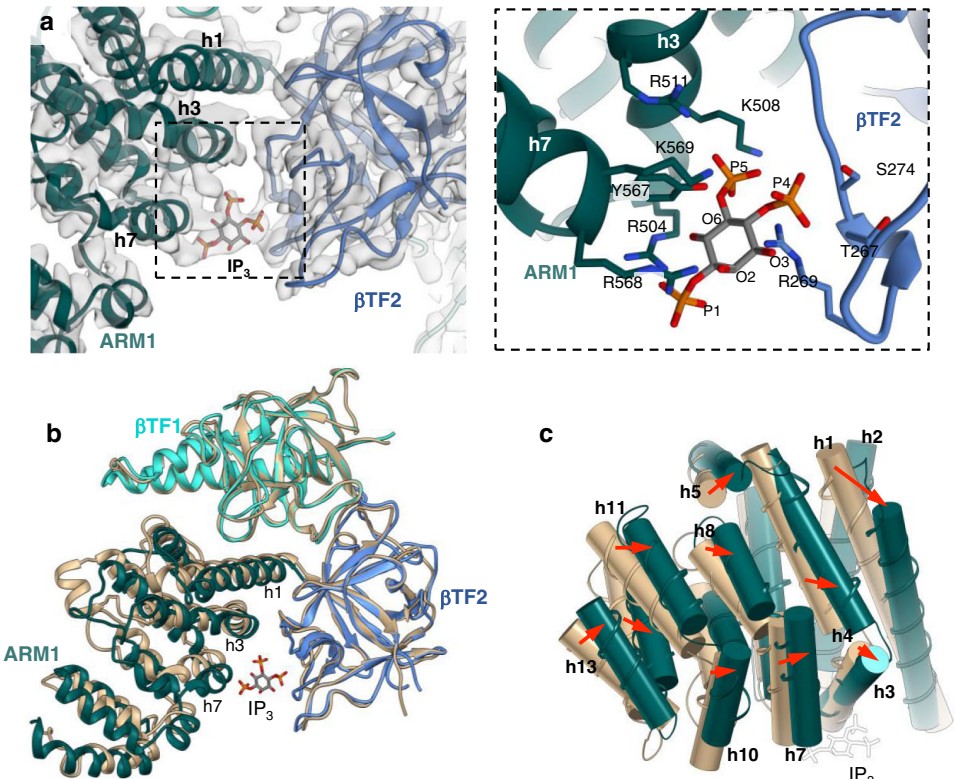

**Fig. 2 | IP₃ induced conformational changes in the ligand binding pocket. a** The IP₃ molecule is fitted to the density bridging βTF2 and ARM1 domains in the CIA-IP₃R1 cryo-EM map overlaid with the corresponding molecular model (left panel). Zoomed-in view of the IP₃ binding pocket structure with coordinating side-chain residues indicated (right panel). **b** Alignment of CIA-IP₃R1 and Ca-IP₃R1 models at the βTF1 domain shows the closure of the IP₃ binding pocket and nearly identical βTF1 and βTF2 backbone structures. **c** ARM1 helices in CIA-IP₃R1 are shifted toward the occupied ligand binding pocket. ARM1 domains (CIA-IP₃R1 colored green; Ca-IP₃R1 colored tan) are shown as thin ribbons overlaid with helices rendered as cylinders and numbered sequentially within the domain. The IP₃ molecule is white.

changes enabling binding of the IP₃ molecule (Fig. 2). It appears that the ARM2 transition between these two distinct conformations correlates with IP₃-binding to the protein. However, mechanistically it remains unclear how IP₃ engages the LBDs and enters its binding pocket - whether it binds by choosing a particular conformation of its binding pocket or pushes LBD conformation into the IP₃-bound complex.

## Structural basis for the IP₃-Ca²⁺ interplay in modulating IP₃R1 activity

The importance of coupling between binding of IP₃ and Ca²⁺ in the regulation of IP₃R gating activity is evident in multiple prior studies[36–38]. However, a mechanistic description of IP₃-Ca²⁺ interplay in regulation of IP₃R activity is still missing. In this study, to understand the structural basis for the modulation of the channel gating with IP₃ and Ca²⁺, we determined IP₃R1 structures under activating conditions in the presence of 2 μM Ca²⁺ (CIA-IP₃R1) and in the presence of 20 μM Ca²⁺ that produces channel inhibition (Ca-IP₃R1). A comparison of the cryo-EM density maps of Ca-IP₃R1 and apo-IP₃R1 revealed strong non-protein densities in Ca-IP₃R1 structure at locations consistent with putative Ca²⁺-binding sites predicted in earlier mutagenesis and functional studies[6,7,24,29]. The ascribed Ca²⁺ densities were validated as described in Methods, establishing the locations of five occupied Ca²⁺-binding sites in each IP₃R1 subunit (Fig. 3). The side chain densities are well resolved in the detected Ca²⁺-binding sites permitting assignment of coordinating residues for the bound Ca²⁺ ions.

Two Ca²⁺-binding sites are identified within the ligand-binding domains across the inter- and intra-subunit interfaces between βTF2 and βTF1 domains. The first site, Ca-I_LBD, is located at an interface between the βTF2 and βTF1' domains from two neighboring subunits.

The Ca²⁺ ion in this site is predominately coordinated by carboxyl groups from D426 and D180 residues in βTF2 and βTF1' domains, respectively. The second site, Ca-II_LBD, is formed across the interface between the βTF2 and βTF1 in the same subunit. This intra-subunit site is composed of two carboxylate oxygen atoms from E283 and backbone carbonyl oxygen atoms from residues K51, K52, F53 and R54 of βTF1 domain (Fig. 3). The locations of these two Ca²⁺ bound ions match well to two predicted Ca²⁺ binding sites in the crystal structure of LBDs[6,29].

Another strong non-protein density is observed in the putative Ca²⁺ sensor region in ARM3 domain (R1582-H2192), which we assigned to a bound Ca²⁺ ion, and this Ca²⁺-binding site is referred to here as Ca-III_S (Fig. 3). This Ca²⁺ ion is stabilized by side chain oxygen atoms from residues E1978 and E2042, Q2045, N1981, and N1971, main chain oxygen from T2654 and a secondary coordination shell may be contributed by H1980 and R1982. Noteworthy, none of the cytosolic Ca²⁺ binding sites described here is similar to that of helix-turn-helix (EF-hand motif) Ca²⁺-binding proteins. Moreover, alignment of 3D structures of the Ca-III_S site in IP₃R1, IP₃R3 and RyR channel confirms a structural conservation of this Ca²⁺ binding site across both families of Ca²⁺ release channels as demonstrated in our previous studies[39]. In our companion study, mutations E1978 or E2042 in the Ca-III_S site markedly affected single channel activity such that altering the negative charge on either aspartate residue, shifted the [Ca²⁺] dependence for activation to the left, consistent with an important role of this site in Ca²⁺ activation of IP₃R1 activity[24]. A conserved E2101 (E4033 in RyR1 and E2005 in IP₃R3) was previously proposed to be responsible for Ca²⁺ sensitivity for activation of IP₃R1 and RyR1 channels[40,41], however, this residue does not directly contribute to coordination of a Ca²⁺ ion in the binding pocket, and

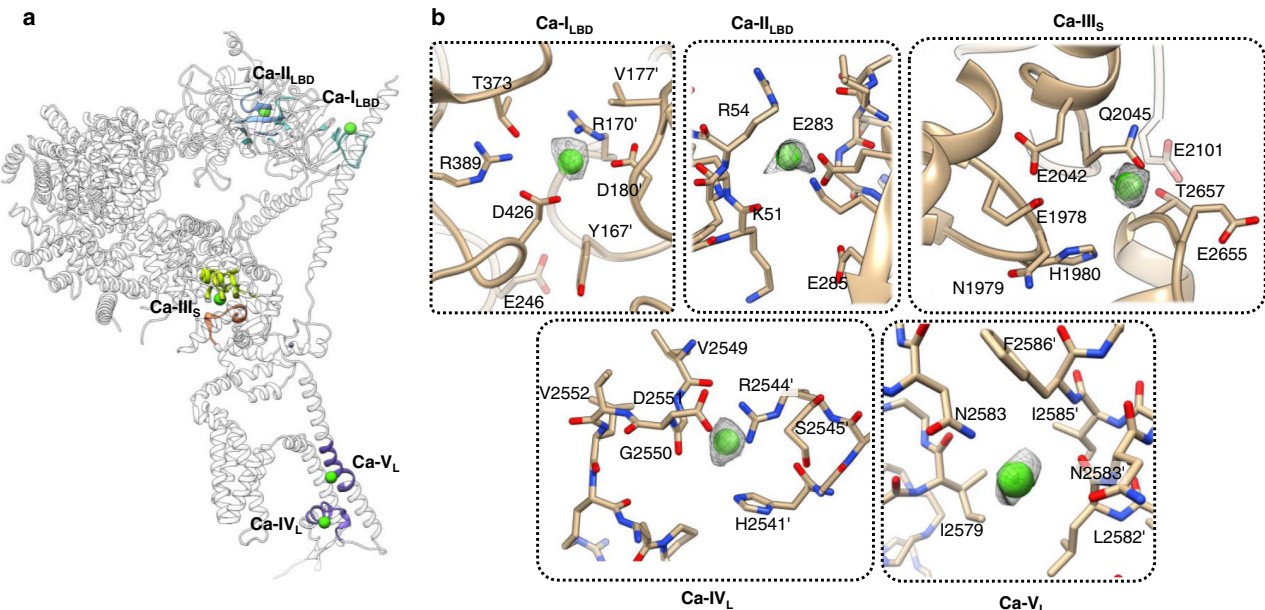

**Fig. 3 | Ca²⁺ binding sites in Ca-IP₃R1. a** Overall topology of Ca²⁺ binding sites is shown for a single subunit of IP₃R1 (white ribbon) with the region surrounding the Ca²⁺ binding site highlighted in the corresponding domain color: Ca-I$_{LBD}$ and Ca-II$_{LBD}$ sites in the ligand binding domains (blue/turquoise); Ca-III$_s$ site in the Ca²⁺ sensor region located within the ARM3 domain (lime green/orange); Ca-IV$_L$ and Ca-V$_L$ sites in luminal vestibule of the TMD (purple). **b** Zoomed-in views of the Ca²⁺ binding sites. Ca²⁺ ions are shown as green spheres and overlaid with corresponding densities displayed at 2-5 σ cutoff values. Residues within 5 Å of the Ca²⁺ ions are displayed in a stick representation and labeled.

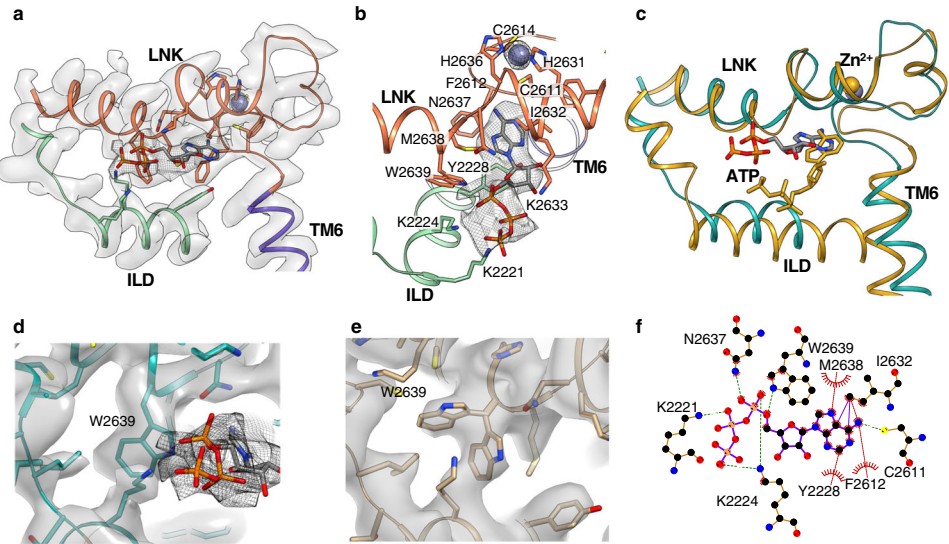

**Fig. 4 | Structural determinants in the ATP binding pocket. a** Cryo-EM densities (transparent gray) are overlaid with a generated model of the ATP binding site identified in the CIA-IP₃R1 structure; the model is colored by domains. The ATP molecule fits to the additional density (gray mesh) observed between the ILD and LNK domains, colored blue-green and orange respectively. **b** Zoomed-in view of the ATP binding pocket with the ATP densities shown in gray mesh; side chain residues within 5 Å of the ATP molecule are displayed. A zinc ion (overlaid with densities at -15σ) is located within the C2H2-like Zn²⁺ finger domain adjacent to the ATP binding pocket. **c** Conserved 3D architecture of the ATP binding pocket in CIA-IP₃R1 (blue-green) and RyR1 (gold; PBD ID: 5TAP); structures are shown as thin ribbon models and ATP molecule shown in sticks. ATP within the RyR1 binding pocket is angled 40° toward the membrane plane with respect to the bound ATP in CIA-IP₃R1. **d, e** Zoomed-in view of cryoEM density maps for CIA-IP₃R1 (**d**) and Ca-IP₃R1 (**e**) overlaid with their respective molecular models; cryo-EM densities corresponding two alternative side chain conformers for W2639 are clearly resolved in Ca-IP₃R1 structure. ATP molecule is fit to the mesh density in '**d**'. **f** Schematic plot of the ATP molecule interacting with surrounding side chains in CIA-IP₃R1 structure. Arcs represent hydrophobic interactions, green dashed lines are H-bonds as calculated in LigPlot+[80].

appears to exert an effect via allosteric interactions with surrounding residues.

By contrast, in the CIA structure, only two cytosolic Ca²⁺ binding sites, Ca-II$_{LBD}$ and Ca-III$_s$, were occupied with Ca²⁺ ions. The structures of the two cytosolic sites are nearly identical to those identified in Ca-

IP₃R1 except for subtle rearrangements in the loop regions (Supplementary Figs. 6, 9). Given that Ca²⁺-occupied Ca-II$_{LBD}$ and Ca-III$_s$ sites are detected at both activating (2 μM) and inhibitory (20 μM) Ca²⁺ concentrations, it would thus appear that Ca²⁺ should bind with higher affinity to these two Ca²⁺-binding sites compared to the Ca-I$_{LBD}$ site,

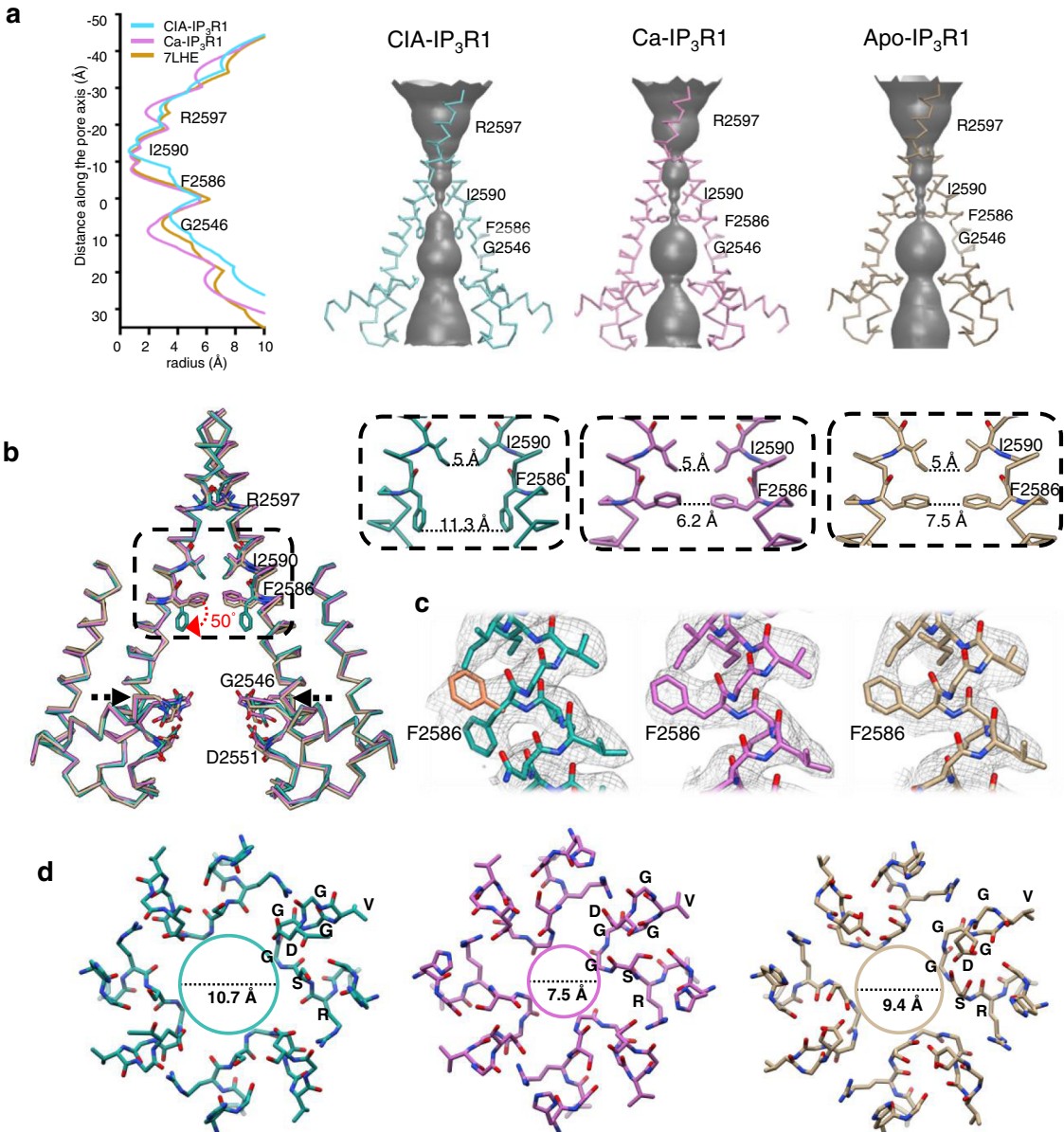

**Fig. 5 | Ion conduction pathway in IP₃R1 upon ligand binding. a** Solvent-accessible pathways in CIA-IP₃R1, Ca-IP₃R1 and apo-IP₃R1. The left panel plots the pore dimensions along the ion conduction pathway and the right panels reveal the solvent accessible volume. **b** The ion conduction pathways in CIA-IP₃R1 (blue-green), Ca-IP₃R1 (pink), apo-IP₃R1 (PDB ID: 7LHE, tan) are aligned and overlaid. The backbone structures for two opposing subunits with several solvent lining side chains are shown and labeled. Red arrow indicates the change in side chain position of F2586 observed in CIA-IP₃R1. Upper right panels show zoomed in views of the area indicated by the box in the left panel. Distances across the ion conduction

pathway at I2590 and F2586 are measured from side-chain atoms from two opposite subunits. **c** Cryo-EM densities overlaid with atomic models for the TM6 gate residue F2586. Two rotamers of F2586 (orange and blue-green) are shown for CIA-IP₃R1. Densities in Ca-IP₃R1 and apo-IP₃R1 permit only one rotamer fit for F2586. **d** Luminal view of the selectivity filter (2544-RSGGGVGD−2551) for CIA-IP₃R1, Ca-IP₃R1, and apo-IP₃R1 at the region indicated by black arrows in 'b' and colored respectively. Distance across the SF is measured between the G2546 Cα atoms from two opposite subunits.

which is occupied only at an inhibitory Ca²⁺ concentration. We found that in the CIA-IP₃R1 structure, the D426 side chain moved away from the Ca-I₍LBD₎ binding pocket and the positively charged R170 shifted toward the location of bound Ca²⁺. This raises the possibility that the conformational differences in Ca-I₍LBD₎ site between Ca-IP₃R1 and CIA-IP₃R1 structures might reflect changes in Ca²⁺-binding affinities. Consistent with our data, mutagenesis of D426 in the Ca-I₍LBD₎ site abolished Ca²⁺ binding[42]. Hence, one could speculate that the Ca-I₍LBD₎ site is a possible candidate for an inhibitory Ca²⁺-binding site, and two other cytosolic Ca²⁺-binding sites (Ca-II₍LBD₎ and Ca-III₍S₎) are activating. As it might have been anticipated, binding of IP₃ "tunes" Ca²⁺-inhibition of the IP₃R channel[36,37].

In both the Ca-IP₃R1 and CIA-IP₃R1 structures, two clear densities, which were interpreted as the bound Ca²⁺ ion, are found in the luminal vestibule leading to the selectivity filter (SF) and the central cavity above the SF. We assigned these densities to bound Ca²⁺ ions (see Methods). One of these Ca²⁺-binding sites, named Ca-IV₍L₎, is shaped by the side chains of residues D2551, G2550, R2544, H2541, S2545 contributed across the interface between two neighboring subunits (Fig. 3). The second Ca²⁺-binding site (Ca-V₍L₎) is located in the central cavity, and Ca²⁺ ion bound at this site is ~2.9 Å away from the amide group in the side chain of N2583 suggesting that the bound Ca²⁺ ion is stabilized in a fully hydrated state. Both identified luminal sites, Ca-IV₍L₎ and Ca-V₍L₎, hint at a high affinity binding site as they are occupied at

relatively low $Ca^{2+}$ concentration. Moreover, the N2583 residue in rodent IP$_3$R1 corresponds to N2543 in humans, and it has been found that the N2543I mutation is closely associated with Gillespie syndrome[43].

## Structural dynamics of the ATP-binding pocket

ATP is known to be a crucial extracellular signaling molecule that increases the open probability of the IP$_3$R channels[13,14,28,44,45]. Previous sequence analysis identified three glycine-rich regions (GXGXXG) in IP$_3$R, reminiscent of the Walker-A motif found in many ATP-binding proteins[46,47], and were presumed to account for the ATP-sensitivity of IP$_3$Rs based on functional studies[10–15]. Although binding of ATP to these motifs has been shown with expressed polypeptides, site-directed mutagenesis and single channel measurements revealed that the Walker-A motifs are not required for ATP modulation of IP$_3$R1[11] or IP$_3$R3[12]. At present, there is a substantial lack of structural knowledge about how ATP regulates the channel gating.

In this quest, using single-particle cryo-EM, we characterize the IP$_3$R1 channel bound to ATP in the presence of its primary activating ligands (CIA-IP$_3$R1). The CIA-IP$_3$R1 structure reveals a previously unseen ATP-binding site located at the interface between the ILD and LNK domains, where clear density is seen (Fig. 4). The ATP molecule fits well into the observed elongated density. Consistent with this assignment, both the apo- and $Ca^{2+}$-bound structures lack this density. The resolution in the ATP-binding site was sufficient to unambiguously identify the pose of the ATP molecule as well as coordinating residues in the binding pocket. This pose allows for the formation of hydrogen bonds between the negatively-charged ATP triphosphate tail and residues K2221, K2224, K2633, N2637, W2639. The residues N2637, M2638, W2639, I2632, C2611, F2612 are responsible for the coordination of the adenine moiety of ATP whereas the residues Y2228, I2632, K2633, N2637, M2638 are involved in the recognition of the hydroxyl groups in the ribose ring of ATP. Therefore, positively charged and hydrophobic residues from ILD and LNK domains coordinate bound ATP molecule within its binding pocket (Supplementary Fig. 8e). The architecture of the ATP-binding pocket is structurally conserved across the intracellular $Ca^{2+}$ release channel family, which includes the ryanodine receptors (RyRs) (Fig. 4c).

Detailed comparisons of the ATP-binding pocket in the apo- or $Ca^{2+}$-bound structures with the CIA-structure reveal that binding ATP induces the most significant change in the side chain conformation of W2639. In the Ca-IP$_3$R1 and apo-IP$_3$R1 structures, we found two clear densities where alternative W2639 rotamers can be fitted (Fig. 4e, Supplementary Fig. 8f). In contrast, only one W2639 side chain conformation was detected in the CIA structure, likely owing to the compatibility of the binding pocket to the ATP molecule (Fig. 4d). Based on these observations, it appears that ATP binding requires flipping of the W2639 side chain inward towards the binding site working as a 'structural switch' for specific ATP binding to the protein.

## Ion conduction pathway

IP$_3$R channels contain a single ion-conducting pore formed at the interface between the TMDs of four subunits along the four-fold axis of the channel. The central pore is lined by the transmembrane helices TM5 and TM6 (Supplementary Fig. 7a). A section of the pore towards the luminal side of the ER membrane, known as the selectivity filter (SF) is shaped by the four luminal P-loops connecting the intervening luminal pore (P) and TM6 helices. The IP$_3$R1 structures described here have a wide luminal vestibule leading to the SF formed by the conserved residues G2546-V2552 from the P-loop at the luminal side of the ion conduction pathway. The diameter of the SF at the entrance to the central hydrophilic cavity of the pore is narrowest in $Ca^{2+}$-bound structure (7.5 Å) and widened to ~9.4 Å and 10.7 Å in apo- and CIA-structures, respectively (Fig. 5). Immediately above the SF there is a water filled cavity, where hydrated $Ca^{2+}$ ions can reside.

Both the Ca- and CIA-structures exhibit two physical sites of constriction in the ion channel pore at the conserved F2586 and I2590 residues of the pore-lining TM6 helix, consistent with previous studies[16,17,20]. The F2586 and I2590 side chains form two hydrophobic seals in the middle of the TM6 helix and near the cytosolic side of the pore, respectively (Fig. 5). Superimposition of IP$_3$R1 structures in apo- versus $Ca^{2+}$-bound and CIA-bound states shows that IP$_3$ binding is associated with coordinated structural changes within the ion conduction pathway. In the $Ca^{2+}$-IP$_3$R1 structure, the cross-sectional area at the pore constrictions has the diameter of 6 Å and 5 Å, respectively, as estimated by the distance between side chains from two opposite subunits at F2586 and I2590. These pore dimensions are similar to that in the nanodisc reconstituted apo-IP$_3$R1 structure, in which the pore is closed, representing non-conducting channel state[20]. In contrast, in the CIA-IP$_3$R1, the diameter of the ion pathway at F2586 constriction is ~11 Å (Fig. 5). This pore expansion is achieved by 50° rotation of the F2586 side chain away from the four-fold axis and down toward the ER lumen. This side chain rearrangement makes the constricted pore permeable to hydrated $Ca^{2+}$ ions, with a diameter of 8–10 Å[48,49]. However, the pore diameter at the position of I2590 remains ~5 Å as seen in the Ca-IP$_3$R1 and apo-IP$_3$R1 structures (Fig. 5a, b). The ion pathway is too narrow at this location for passage of hydrated $Ca^{2+}$. Consistent with this observation, the pore in structures of several $K^+$ channels and the human CLC-1 chloride channel are also not wider than a hydrated ion under activating conditions[50–53]. Thus, it is conceivable that a slightly narrow tunnel lined with dynamic side chains apparently can provide a good passage for hydrated ions.

Furthermore, the pore-lining TM6 helix in the CIA- and $Ca^{2+}$-bound structures of IP$_3$R1 adopts the π-helix conformation at residues M2576-V2581 residue and confers flexibility to the ion conduction helix bundle (Supplementary Fig. 7c). By contrast, the apo-structure of ND-reconstituted IP$_3$R1 exhibits straight TM6 helices at this region[20]. On the basis of these observations, we propose that gating in IP$_3$R1 involves an α-to-π secondary structure transition.

In both the Ca-IP$_3$R1 and CIA-IP$_3$R1 structures, the SF and a central cavity are continuous with the luminal solvent, and hydrated $Ca^{2+}$ ions can diffuse into and out of the water filled luminal vestibule of the ion conduction pathway (Fig. 5b, d). Thus, the Ca-V$_L$ site is easily accessible for binding of $Ca^{2+}$. However, it seems that the occupancy of this $Ca^{2+}$ binding site is dependent on conformation of the TM6 helix. The TM6 rearrangement upon the channel activation is accompanied by movement of the F2586 side chain in the hydrophobic constriction and shift of the N2583 side chain, which coordinates $Ca^{2+}$ ion in the Ca-V$_L$ site (Fig. 3; Supplementary Fig. 9). This can potentially weaken $Ca^{2+}$-coordinating interactions or even force loss of $Ca^{2+}$ into the water filled central cavity from where it can travel down the electrochemical gradient in vivo.

Overall, the conformation of the ion conduction pore in the CIA-IP$_3$R1 structure is compatible with that in the IP$_3$R1 channel characterized in the presence of adenophostin A and $Ca^{2+}$, [17]. We note, however, an additional strong density at the location of F2586 in the pore-lining TM6 helix, where the F2586 side chain can be alternatively docked in the CIA-IP$_3$R1 structure. This density points to the central 4-fold axis of the pore and is consistent with the conformation of the F2586 side chain observed in the apo- or Ca-IP$_3$R1 structures representing the IP$_3$R channel in closed (non-conducting) state (Fig. 5b, c). In addition, the structural rearrangements of the SF observed in CIA-IP$_3$R1 structure stand in contrast to what is seen in the IP$_3$R1 visualized in the presence of activating concentrations of adenophostin A and $Ca^{2+}$ (AdA-IP$_3$R1), which SF is narrower (~8.5 Å) than that in apo-IP$_3$R1[17]. These structural differences might reflect the separate contexts in which these structures were determined, CIA-bound in a nanodisc and the other in the detergent. Noteworthy, the narrow diameter of the SF in apo-IP$_3$R1 likely does not reflect any physiologically relevant conformation as the luminal vestibule of the channel pore is normally

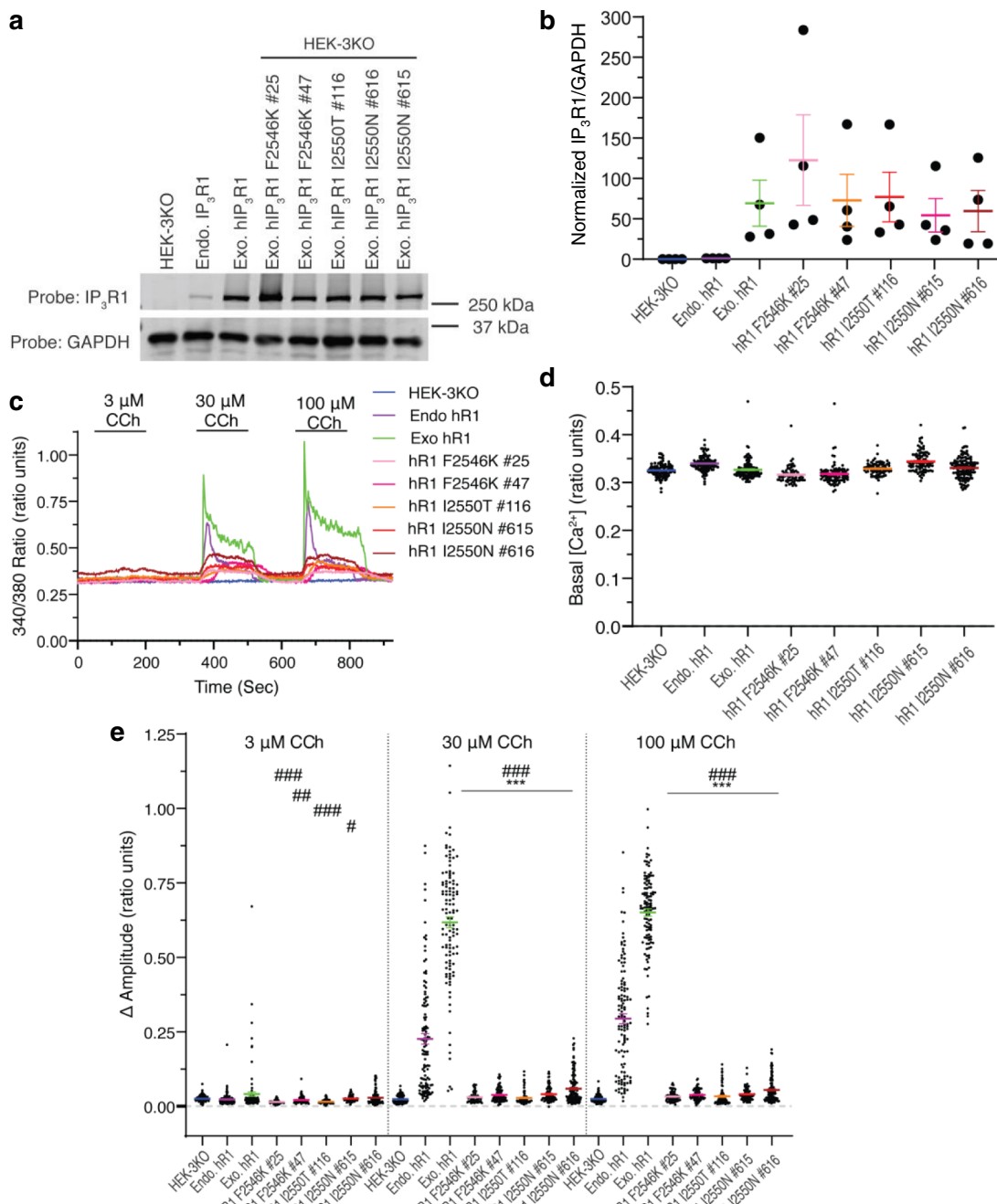

**Fig. 6 | Functional validation of gating residues in ion conduction pathway.**
**a** Immunoblots of lysates prepared from the indicated HEK cell lines. HEK-3KO and endogenous hIP$_3$R1 (Endo. hIP$_3$R1) were generated by CRISPR/Cas 9 technology, the former is null for all IP$_3$R subtypes while the latter expresses only IP$_3$R1. Mutations and exogenously expressed IP$_3$R1 (Exo. hR1) were stably expressed in HEK-3KO cells. All stable clonal cell lines result in expression levels above that of Endo. hIP$_3$R1, as quantified in **b**. Data are presented as mean values +/- SEM ($n$ = 4 independent experiments). HEK cell lines are color coded: blue−HEK-3KO, purple−endo hIP$_3$R1, green−exo hIP$_3$R1, pink−hIP$_3$R1 F2546K #25, magenta−hIP$_3$R1 F2546K #47, orange−hIP$_3$R1 F2550T #116, red−hIP$_3$R1 F2550N #615. brown−hIP$_3$R1 F2550N #616. **c** Representative single cell Ca$^{2+}$ traces in fura-2 loaded HEK cell lines stimulated with increasing concentrations of the muscarinic agonist carbachol (CCh). **d** Basal fluorescence, prior to stimulation in HEK cell lines. **e** Pooled data depicting the response of cell lines to CCh stimulation. Data in d-e are presented as mean values +/- SEM ($n$ = at least 60 cells over three independent experiments). *significantly different from Endo. hIP$_3$R1. #significantly different from Exo. hIP$_3$R1. For 3 μM CCh: ###$p < 0.0001$; ##$p = 0.0007$; #$p = 0.0395$; for 30 μM CCh and 100 μM CCh: ###***$p < 0.0001$ One-way ANOVA with Tukey's post-hoc test. Source data are provided as Source Data file.

exposed to the ER lumen containing calcium in the low millimolar range[54]. While the Ca$^{2+}$ binding to the luminal sites may have a regulatory function[55,56], it remains to be established whether Ca$^{2+}$ store depletion could lead to dissociation of Ca$^{2+}$ from the luminal vestibule.

To further investigate functionality of the amino acids at the constriction site, mutants of residues F2586 (F2586K) and I2590 (I2590N and I2550T) were generated in human IP$_3$R1 (hR1). The F2586K

mutation replaces a large, hydrophobic Phe with a positively charged Lys, while the I2590N and I2550T mutations are associated with human disease[43,57,58]. The mutations were stably overexpressed in cells engineered via CRISPR/Cas9 to lack all three endogenous IP$_3$Rs (HEK-3KO)[5] (Fig. 6a, b). All three mutations resulted in significantly attenuated agonist-induced Ca$^{2+}$ release (Fig. 6c, e) when compared with agonist-stimulated release in cells expressing human IP$_3$R1 either at

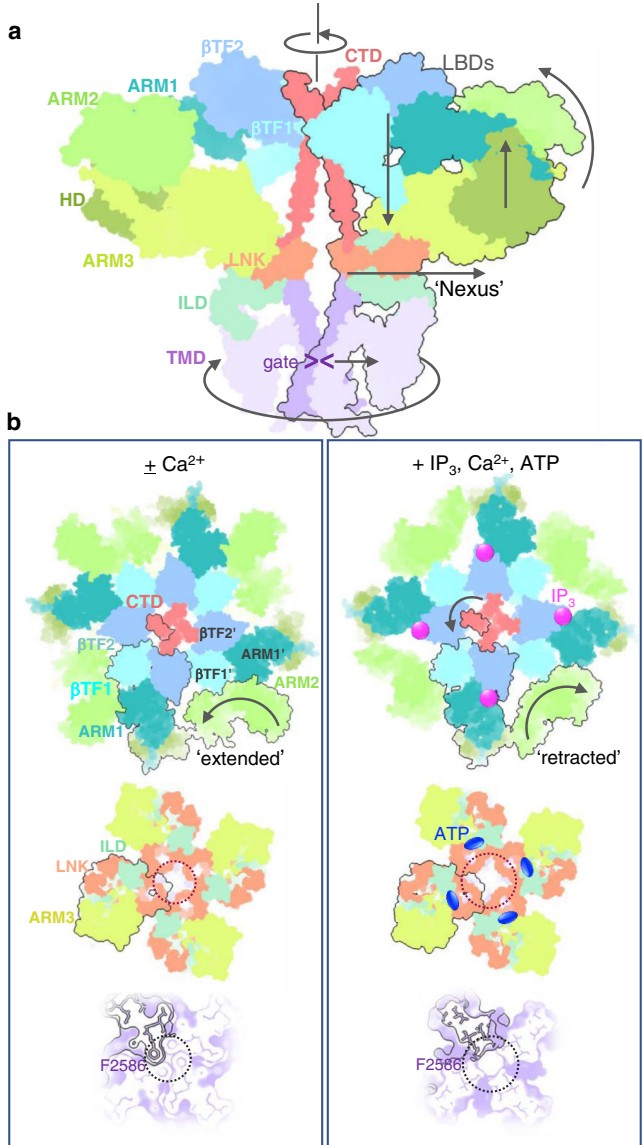

**Fig. 7 | Schematics of ligand-induced structural changes underlying activation of IP₃R1 channel. a** A conformational wave generated upon binding of the activating ligands propagates from the LBDs forming the apical portion of the channel via the ILD/LNK assembly ('nexus') towards the channel pore. Depicted are two opposing subunits colored by domains. Domain motions are indicated with arrows. **b** Conformational changes underlying binding of IP₃, Ca²⁺ and ATP. Intrinsic flexibility of ARM2 domain allows for a reversible ratcheting mechanism where ARM2 switches between 'extended' and 'retracted' conformations. The extended conformation is restrictive for binding of IP₃ (top left), while 'extended' conformation is suitable for capturing IP₃ due to release of structural constraints at interfaces between ARM2 and βTF1' and ARM1' from the neighboring subunit (top right). Allosteric nexus comprising LNK and ILD domains (middle panels) and the channel pore at F2586 (bottom panels) are expanded in the presence of activating ligands. The domains in Ca-IP₃R1 (left column) and CIA-IP₃R1 (right column) structures are viewed along the central 4-fold axis from the cytosol with one subunit outlined in black.

endogenous or overexpressed levels. The basal [Ca²⁺] was not altered in any of the mutations (Fig. 6c, d). Replacing F2586 with the positively charged K creates electrostatically unfavorable environment for the permeating Ca²⁺ via the open pore (Ca²⁺ blockage). The presence of neutral T or hydrophilic N at the I2590 position is clearly not favorable for inter-subunit hydrophobic side-chain packing that will perturb the pore stability thereby affecting the channel open probability. Thus, our

results confirm the critical nature of the F2586 and I2590 residues in IP₃R1 channel function.

## Discussion

The high-resolution structures of Ca²⁺- and Ca²⁺/IP₃/ATP-bound IP₃R1 presented in this study highlight the correlation between the protein dynamics and its propensity for accepting multiple cellular signals that regulate the channel activities, such as ligand-binding, gating and regulation. Considering the remote distances between the cytoplasmic ligand-binding sites and ion conduction pathway in the channel, the intrinsically flexible 3D architecture of the IP₃R provides the basic premise behind the allosteric regulation of the channel activity that involves the transfer of ligand-evoked signals from the distinct ligand-binding domains to the ion conducting channel gate (Fig. 7). This process is mediated through molecular interactions at inter- and intra-domain interfaces embedded in the conformational landscape of the protein. While some structural features of the ligand-bound IP₃R1 that we highlighted here confirm our previous observations[16,17], the present study reveals mechanisms for the conformational transitions underlying IP₃R1 activity.

Using deep-learning based modeling[22] and 3D variability analysis[23], we observe that the ARM2 domain exhibits the unique motions within the dynamic conformational ensemble, resulting in two predominant conformations seen in CIA- and Ca²⁺- bound states. In the IP₃-binding process in the context of the tetrameric channel assembly, intrinsic flexibility of the ARM2 domain is highly important as its dynamic behavior perhaps allows to release structural constraints imposed at interaction interfaces between LBD and ARM2 domains that facilitates the conformational selection from a pre-existing ensemble of IP₃-binding pocket conformations and promotes specific binding of the IP₃ molecule. Based on our results, the dynamic interactions between the ARM2 and LBDs can be envisioned as 'a reversible ratchet' mechanism with the gear (ARM2) held in place by its contacts with βTF1 and ARM1 domains from the neighboring subunit (Fig. 7a). This dynamic re-arrangement allows direct (clockwise) as well as reverse (counterclockwise) movements of the ARM2 between 'extended' and 'retracted' positions with the latter favorable for specific binding of IP₃. Altogether, these results allow us to delineate distinct steps in IP₃-binding conformational cycling, including interfacial rearrangements. It appears that IP₃ may work by tuning the Ca²⁺ sensitivity of IP₃R1 by decreasing the Ca²⁺ affinity of the Ca-I_LBD site and stimulating Ca²⁺ binding to the Ca-II_LBD site. The conformational changes observed in the IP₃-binding domains unequivocally propagate through other cytoplasmic domains to the TMDs that form the channel pore (Fig. 7).

It is important to note that our high-resolution CIA-IP₃R1 structure demonstrates that under saturating activating conditions used in this study, the gating state of the channel pore likely relies on conformational dynamics of the F2586 side chains that can potentially alter the relative stability of conformational states of the pore. Hence, instead of directly going from closed to open conformation, the pore-forming elements can adopt a mixture of conformations resulting in different levels of IP₃R1 channel activity observed under constant experimental conditions in electrophysiological studies[59,60]. This phenomenon known as modal gating has been observed in many other ion channels including RyR Ca²⁺ release channel[61–64].

The observed degree of pore opening at I2590 in the CIA-IP₃R1 structure is not as wide as a hydrated Ca ion, however it might provide a good passage for Ca²⁺ permeation using a mechanism relying on dynamic side chains shaping the ion conduction pathway. Noteworthy, the structures of many other ion channels studied under activating conditions exhibit the pore opening to less degree than a hydrated ion[50–53].

The presence of the π-gating hinge in IP₃R1 channel is consistent with structures of IP₃R3 channel, solved by cryo-EM[18,21], suggesting that

gating in both channel types is prone to α-to-π transitions. However, the detergent-solubilized IP$_3$R3 channel exhibits the π-helix in TM6 in the non-conducting (closed) state, which transitions to an α-helix in the presence of IP$_3$ and Ca$^{2+}$. Strikingly, unlike IP$_3$R3, the ligand-bound structures of IP$_3$R1 reconstituted into lipid nanodisc solved in this study, appear to favor the π-helix upon the channel activation. Together, these data suggest that the lipid environment plays an essential role in formation and maintenance of π-helix structure.

Furthermore, the results of the present study are supportive of the importance of side chain dynamics in ATP binding by IP$_3$R1 protein. Studies suggest that the three IP$_3$R isoforms exhibit different sensitivity to ATP with IP$_3$R1 being enhanced by micromolar ATP and IP$_3$R3 being augmented by millimolar ATP, whereas IP$_3$R2 is more sensitive to ATP but only at sub-micromolar IP$_3$[12,65,66]. Our analysis revealed a side chain conformational preference for W2639 that contributes to ATP binding. This observation supports the hypothesis that ATP-binding specificity of IP$_3$Rs might be mediated by the recognition of specific rotameric states of key residues in the ATP-binding pocket. Additionally, the presence of conserved lysine residues responsible for the coordination of ATP phosphates in IP$_3$R1 likely contribute to isoform specific ATP binding affinity. Decoding the structural basis of exquisite isoform specificity with respect to a modulatory role of ATP in the IP$_3$R family will certainly be critical in future studies.

Overall, our analysis exposes a structural mechanism for the susceptibility of IP$_3$R1 to binding of IP$_3$, based on the conformational selection of the ligand-binding pocket, which may adapt different conformations in its unbound state for which IP$_3$ binds selectively to one of these conformations. We propose that IP$_3$/Ca$^{2+}$-evoked gating involves 'a conformational wave' that unequivocally propagates via the cytoplasmic domains towards the pore-forming TMDs (Fig. 7 and Supplementary Movies 1–3). The present results highlight a key role of specific interactions in the side chain network surrounding the ion conduction path in regulating gating behavior of IP$_3$R channels. Notably, the structural rearrangements in the cytoplasmic domains are substantial relative to the transmembrane domain suggesting that the structural flexibility allows the protein domains to adapt to their individual molecular binding partners facilitating the binding process. Our study provides a firm structural framework for interpreting large amounts of functional and biochemical data accumulated during more than three decades of research targeting IP$_3$-mediated Ca$^{2+}$ release. However, further systematic work combining structural analysis with site-directed mutagenesis and electrophysiological characterization is still needed to test connections between functional paths and protein motions underlying ligand-mediated allosteric information transfer within the channel.

## Methods

### IP$_3$R1 purification and reconstitution into nanodiscs

Purification of neuronal IP$_3$R1 reconstituted was performed as previously described[20,67]. Briefly, rat cerebellar membranes were solubilized in 2 mM Lauryl Maltose Neopentyl Glycol (LMNG, Anatrace) and 0.1% (w/v) l-α-phosphatidylcholine (PC, Sigma) for 2 h at 4 °C. Non-solubilized material were cleared by ultracentrifugation (100,000 x $g$) and the supernatant was applied to an immunoaffinity matrix pre-coupled with monoclonal antibodies (10 mg; T433 epitope of IP$_3$R1) in 50 mM Tris-HCl (pH 7.4), 150 mM NaCl, 1 mM DTT, 1 mM EDTA, 0.02 mM LMNG. Reconstitution of IP$_3$Rs into lipid-nanodiscs occurred "on-column" as we described in ref. 20: 1.2 mM POPC (1-palmitoyl-2-oleoyl-glycero-3-phosphocholine, Avanti Polar Lipids) prepared in 3% DDM (n-dodecyl-β-d-maltoside, Avanti Polar Lipids, Inc), 50 mM Tris-HCl (pH 7.4), 150 mM NaCl, 1 mM DTT, 1 mM EDTA was added to the IP$_3$R1-bound immunoaffinity column matrix followed by the addition of 0.6 mg/ml MSP1E3D1 (Cube Biotech). Detergent removal was achieved by the addition of 0.25 mg/ml activated Bio-Beads-SM2

(BioRad) for 16 h at 4 °C. IP$_3$R1 was eluted with 500 μM of T433 peptide and concentrated using a 100 kDa Amicon centricon (Millipore).

### CryoEM sample preparation and data acquisition

The nanodisc reconstituted purified IP$_3$R1 (2 mg/ml) was vitrified after 30 min in incubation with 20 μM free Ca$^{2+}$ to obtain the structure of IP$_3$R1 bound with Ca$^{2+}$ (Ca-IP$_3$R1) or with 2 μM free Ca$^{2+}$, 10uM of IP$_3$ and 1 mM ATP to obtain Ca/IP$_3$/ATP-bound IP$_3$R1 (CIA-IP$_3$R1). Vitrification of the IP$_3$R1 samples was performed using a Vitrobot Mark IV (ThermoFisher Scientific, Inc.) as described earlier[20]. Blotting was done with filter paper Whatman 542 containing 8–14 μg Ca/g paper. The final free Ca$^{2+}$ concentrations were determined using MaxChelator (http://maxchelator.stanford.edu/oprog.htm). CryoEM data acquisition was performed with a Titan Krios microscope (ThermoFisher Scientific, Inc.) operated at 300 keV, equipped with a BioQuantum energy filter (Gatan, Inc.) with zero-loss energy selection slit set to 20 eV. All data sets were acquired using the EPU software (ThermoFisher Scientific, Inc.) at a nominal TEM magnification of 130,000X and recorded on a K2 Summit direct electron detector (Gatan, Inc.) operated in super-resolution counting mode with a calibrated physical pixel size of 1.07 Å. The data acquisition parameters (total movie stacks, dose rate, total dose, defocus range, etc.) are summarized in Table 1 and Supplementary Figs. 1, 3, 5.

### Image processing

Movie stacks were motion corrected with dose weighting and binned 2 × 2 by Fourier cropping resulting in a pixel size of 1.07 Å using MotionCor2[68]. The motion-corrected micrographs were evaluated by 'e2evalimage.py' in EMAN2[69] and were used for all the following image processing. Contrast transfer function (CTF) determination was performed using Gctf[70]. Particles were picked using the NeuralNet autopicking procedure implemented in EMAN2 and the .box files were imported to RELION3[71] and were subjected to iterative 2D and 3D classification. Our previously resolved cryo-EM map of apo-IP$_3$R1 (Electron Microscopy Data Bank (EMDB: EMD-23337) was low-pass filtered to 60 Å as a starting reference map. After 3D auto-refinement, two overall 3D reconstructions at 3.50 Å and 3.26 Å resolution (using FSC at 0.143[72]) were generated for the CIA-IP$_3$R1 and Ca-IP$_3$R1 respectively (Supplementary Figs. 2, 4). Local resolution estimation was performed by Resmap[73], the results show ARM2 domain is the most flexible region of the IP$_3$R1 map, which also was observed in our published apo-IP$_3$R1 structure (EMDB: EMD-23337)[20]. To improve the map quality of ARM2 domain for all three states (CIA-IP$_3$R1, Ca-IP$_3$R1 and apo-IP$_3$R1), we performed symmetry expansion by running the command "relion_particle_symmetry_expand" with C4 symmetry, followed by multiple rounds of 3D classification with a mask including only ARM2 domain, and the particles from the best featured classes were subjected to local refinement (Supplementary Figs. 1–5). The expanded consensus particles were imported to EMAN2[22,69] and cryoSPARC[23,74] to analyze the conformational heterogeneity of the particles. Flow charts for data processing steps are presented in Supplementary Figs. 1, 3, 5.

### Gaussian Mixture Model (GMM) based heterogeneity analysis in EMAN2

Overall analysis was performed as described in[22], with adaptations to combine the three particle populations and C4 pseudosymmetry. Particles of IP$_3$R1 under three different conditions (apo, CIA and Ca) were combined and a single averaged structure with C4 symmetry was reconstructed from the particles, which was used as the neutral state structure for GMM based heterogeneity analysis. To analyze the movement of each subunit independently, each IP$_3$R1 particle was duplicated four times, each time with orientation assigned to a different asymmetrical unit. During the heterogeneity analysis, the model was constrained such that Gaussian functions corresponding to the

ARM2 domain could only translate/rotate together as a rigid body, and only a single subunit of IP$_3$R1 was allowed to deviate from the neutral state position. After training the network model, the first eigen-vector on the conformational space exhibited a clear extended-retracted motion of the ARM2 domain. The three distinct particle subsets all exhibited motion on the same pathway in the ARM2 domain, but to differing degrees, with the CIA data set exhibiting more motion than the other two states.

### 3D variability analysis in cryoSPARC

Symmetry expanded consensus particles were analyzed using the 3D Variability Analysis (3DVA) module[23]. For each cryo-EM data set, five modes of variability were calculated. Ten maps for each mode were generated using a 4 Å low-pass filter. In each state, the ARM2 domain displayed the largest motions of similar trajectories, regardless of symmetric and symmetry-breaking variability. To visualize the conformational variability across the channel protein assembly, the maps from the modes representing C4 motion (mode 0 of CIA-IP$_3$R1, mode 4 of Ca-IP$_3$R1 and mode 1 of apo-IP$_3$R1) were examined in UCSF Chimera[75] as volume series and recorded as movies (Supplementary Movies 1–3).

### Model building

We utilized a consistent approach for protein model building in both CIA-IP$_3$R1 and Ca-IP$_3$R1 cryo-EM density maps. Our previously published cryo-EM structure of apo-IP$_3$R1 in nanodisc (PDB ID: 7LHE;) was used for rigid body fitting in UCSF Chimera[75] followed by an initial round of flexible fitting of the IP$_3$R1 subunit in COOT. Density modification "phenix.resolve_cryo_em" was performed on the half-maps resulting in improved resolvability in the density maps allowing for further model optimization. AlphaFold[76] was used to generate optimized models for each of the ten domains within an IP$_3$R1 subunit using the initial IP$_3$R1 model for each domain as a template. The results were then concatenated and tetramerized in Chimera, refined against the density map with "phenix.real_space_refine" with default options and manual optimization in Coot to maximize fit to density, minimize Ramachandran angle outliers and eliminate steric clashes. Ligands, including lipids, calcium, IP$_3$ and ATP, were identified using "pw_ligands.py", a stand-alone ligand identification tool based on Pathwalking[77]. Briefly, "pw_ligands.py" identifies all non-protein density, analyzes the geometry of the un-modeled density and masks out ligands and water/ion density from the cryoEM map. At the reported resolutions, identification of non-ion/non-water ligands is robust, though it is difficult to distinguish waters from ions based purely on density profile. All prospective water/ion densities were examined; only clearly distinguishable densities with obvious coordinating chemistry were considered as ions. The ligands were fit to the masked densities and refined along with the protein model using "phenix.real_space_refine". Model validation was carried out using EMRinger and MolProbity in PHENIX. Map-model visualization was performed in Coot and UCSF Chimera. Interfaces described in the paper were identified with PDBePisa[78], ChimeraX[79] LigPlot+[80] and HOLE[81]. Figures and movie were prepared using UCSF Chimera; ChimeraX and VMD 1.9.4[82].

### Generation of IP$_3$R mutants

A two-step QuikChange mutagenesis protocol was used to introduce amino acid substitutions into cDNA encoding the human IP$_3$R1 (hIP$_3$R1; NM_001099952 in pDNA3.1 (Obtained from Annetta Wronska at Columbia University)). Mutagenesis and all DNA modifications were carried out using *Pfu* Ultra II Hotstart 2X Master Mix (Agilent). Mutagenesis primers for hIP$_3$R1 F2586K forward (TCTTCATGGTCATCATCA TTGTTCTTAACCTGATTAAGGGGGTTATCATTGACACT), hIP$_3$R1 F258 6K reverse (AGTGTCAATGATAACCCCCTTAATCAGGTTAAGAACAAT GATGATGACCATGAAGA), hIP$_3$R1 I2590N forward (TTGGGGTTATCA ATGACACTTTTGCTGACCTACGTAGTGAGAAGCAG), hIP$_3$R1 I2590N

reverse (CTGCTTCTCACTACGTAGGTCAGCAAAAGTGTCATTGATAA CCCCAA), hIP$_3$R1 I2590T forward (TTGGGGTTATCACTGACACTTTT GCTGACCTACGTAGTGAGAAGCAG), and hIP$_3$R1 I2590T reverse (CT GCTTCTCACTACGTAGGTCAGCAAAAGTGTCAGTGATAACCCCAA), were synthesized by Integrated DNA Technologies (IDT). In addition to encoding the specified mutations, primers also silently introduced a restriction site for verification purposes. The coding regions for all constructs were confirmed by sequencing.

### Cell culture and generation of HEK cell lines stably expressing IP$_3$R monomers

HEK-293s were originally obtained from ATCC (CRL-3022). HEK-3KO cells, HEK293 cells engineered through CRISPR/Cas9 for the deletion of the three endogenous IP$_3$R isoforms[83] and HEK293 cells modified by CRISPR/Cas9 to only express endogenous hIP$_3$R1 (Endo. hR1) were grown at 37 ˚C with 5% CO$_2$ in Dulbecco's Modified Eagle Medium (DMEM) supplemented with 10% fetal bovine serum, 100 U/ml penicillin, 100 µg/ml streptomycin (Gibco/Life Technologies). Transfection of HEK-3KO cells to exogenously express desired hIP$_3$R1 WT or mutant constructs stably was performed as previously described[83]. In brief, five million cells were pelleted, washed once with PBS, and resuspended in either Nucleofector Solution T (Lonza Laboratories) or a homemade transfection reagent (362.88 mM ATP-disodium salt, 590.26 mM MgCl$_2$ 6.H$_2$O, 146.97 mM KH$_2$PO$_4$, 23.81 mM NaHCO$_3$, and 3.7 mM glucose at pH 7.4). 4–6 µg of DNA was mixed with the resuspended cells and electroporated using the Amaxa cell nucleofector (Lonza Laboratories) program Q-001. Cells were allowed to recover for 48 h before passage into new 10 cm plates containing DMEM media supplemented with 2 mg/ml Geneticin sulfate (G418; VWR). Following 7 days of selection, cell colonies were either picked and transferred to new 24-well plates or diluted into 96 well-plates, both of which contained DMEM media supplemented with 2 mg/ml G418. Diluted wells in the 96-well plate were screened after 7 days for the presence of a single colony of cells growing in each well. Those wells that exhibited multiple colonies growing were excluded from further consideration. 10–14 days after transfection, wells–both 96 and 24–that exhibited growth were expanded and those expressing the desired constructs were confirmed by western blotting.

### Cell Lysis and SDS-PAGE analyses

HEK-3KO cells and HEK-3KO cells stably expressing IP$_3$R constructs were harvested by centrifugation (200 x $g$ for 5 min), washed once with PBS, and solubilized in membrane-bound extraction lysis buffer containing 10 mM Tris-HCl, 10 mM NaCl, 1 mM EGTA, 1 mM EDTA, 1 mM NaF, 20 mM Na$_4$P$_2$O$_7$, 2 mM Na$_3$VO$_4$, 1% Triton X-100 (v/v), 0.5% sodium deoxycholate (w/v), and 10% glycerol supplemented with a cocktail of protease inhibitors (Roche, USA). Lysates were incubated for a minimum 30 min on ice and cleared by centrifugation (16,000 x $g$ for 10 min) at 4 °C. Protein concentrations in cleared lysates were determined using D$_c$ protein assay kit (Bio-Rad) and 4x SDS gel loading buffer was subsequently added to 5 µg of lysate. Proteins were resolved using 8% SDS-PAGE and subsequently transferred to a nitrocellulose membrane (Pall Corporation). Membranes were probed with a rabbit polyclonal antibody against the C-terminal 19aa of IP$_3$R1 (custom generated by Antibody Research Corporation) at a 1:1000 dilution, GAPDH (#AM4300, Invitrogen) at a 1:75,000 dilution, and the appropriate Dylight™ 800CW secondary antibodies at a 1:10,000 dilution (SA535571 and SA535521; Invitrogen). Membranes were imaged with an Odyssey infrared imaging system and quantified using Image Studio Lite (LICOR Biosciences).

### Single Cells Ca$^{2+}$ imaging: measurement of cytosolic Ca$^{2+}$ in intact cells

Single cell Ca$^{2+}$ imaging was performed in intact cells as described previously[83]. Glass coverslips were plated with HEK-3KO, Endo. hR1,

Exo. hR1, or HEK-3KO cells stably expressing IP$_3$R mutant constructs at least 18 h prior to imaging experiments. Subsequently, the glass coverslips were mounted onto a Warner chamber and the cells were loaded with 2 μM Fura-2/AM (Molecular Probes) in Ca$^{2+}$ Imaging Buffer (Ca$^{2+}$ IB;10 mM HEPES, 1.26 mM Ca$^{2+}$, 137 mM NaCl, 4.7 mM KCl, 5.5 mM glucose, 1 mM Na$_2$HPO$_4$, 0.56 mM MgCl$_2$, at pH 7.4) at room temperature for 25 min. Following loading, cells were then perfused with Ca$^{2+}$ IB which provided a basal 340/380 ratio of the [Ca$^{2+}$] and stimulated with 3 μM, 30 μM, and 100 μM carbachol (CCh) to obtain measurements of agonist-induced Ca$^{2+}$ release into the cytoplasm.

Ca$^{2+}$ imaging was performed using an inverted epifluorescence Nikon microscope with a 40x oil immersion objective. Cells were alternately excited at 340 and 380 nm, and emission was monitored at 505 nm. Images were captured every second with an exposure of 15 ms and 4 × 4 binning using a digital camera driven by TILL Photonics software. Image acquisition was performed using TILLvisION software and data was exported to Microsoft Excel where the means were calculated. Statistical analysis of at least $n = 3$ experiments for each cell line was performed in GraphPad Prism (one-way ANOVA with Tukey's test).

### Reporting summary

Further information on research design is available in the Nature Portfolio Reporting Summary linked to this article.

## Data availability

Data supporting the findings of this manuscript are available from the corresponding author upon reasonable request. The cryo-EM density maps have been deposited in the Electron Microscopy Data Bank (EMDB) under accession codes EMD-27983 (CIA-IP$_3$R1), EMD-27982 (Ca-IP$_3$R1). The coordinates have been in the RCSB Protein Data Bank (PDB) under accession codes 8EAR (CIA-IP$_3$R1), 8EAQ (Ca-IP3R1). Data from our previously resolved cryo-EM map of apo-IP$_3$R1 (EMD-23337 and coordinates 7LHE were used in this study. Source data underlying Fig. 6 is available as a Source Data file. Source data are provided with this paper.

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

## Acknowledgements

We thank Venkata Mallampalli at CryoEM Core Facility at UTHealth at Houston for technical support on cryo-EM. This work was supported by grants from the National Institutes of Health (R01GM072804 and 3R01GM072804-11S to I.I.S.; DE019245 to D.I.Y.; R01GM080139 to S.J.L.), American Heart Association grant (18CDA34110086 to M.R.B.) and Welch Foundation research grants (AU-2014-20190330 and AU-2014-20220331 to I.I.S.). TFS Titan Krios Equipment subsidized by CPRIT Core Facility Award RP190602 was used for Cryo-EM imaging in this work.

## Author contributions

I.I.S. and D.I.Y. conceived the project; G.F., M.R.B., A.B.S. prepared samples of IP$_3$R1; G.F. collected cryo-EM data and determined structures; G.F., M.R.B. and M.L.B. built and refined the atomic models; G.F., M.R.B., M.L.B. and I.I.S. analyzed the structures; S.J.L., M.C. and G.F performed variability analysis; L.E.T and V.A made mutations in IP$_3$R and performed functional analyses of IP$_3$R activity; I.I.S. wrote the paper with the help from M.R.B. and G.F., and all other co-authors.

## Competing interests

The authors declare no competing interests.
