## [Peer Review File · Nature Communications]

Conformational motions and ligand-binding underlying gating and regulation in IP3R channelReviewers' Comments:

Reviewer #1:

Remarks to the Author:

In this manuscript Fan et al. determined the near-atomic structure (3.5 and 3.26 Å) of IP3R1 in the presence of activating Ca+IP3+ATP and in the presence of inhibiting Ca. They localized the ATP, IP3, and Ca binding sites, with good agreement with existing near-atomic 3D structures of the IP3R-3 isoform (by the Hite and Karakas groups). Additionally, they identified new putative Ca binding sites and for the IP3R-Ca dataset, they also potentially found an inhibitory site for Ca. Some of the sites were validated using functional experiments. Variability analysis identified two conformations of the ARM2 domain, "extended" and "retracted", where an effect of IP3 is to increase the amplitude of the movement of the ARM2 domain.

MAJOR ISSUES

A major problem is the disconnect between the title/abstract and the rest of the manuscript. The application of the deep neural network method, developed by the team, is not obvious in the manuscript, and the dynamic aspect is quite scant. As such, the manuscript focuses mainly on the different binding sites and on the structure of the pore.

MINOR ISSUES

Line 24: please spell out GMM

Line 154: "each individual 2-D particle "

The term is confusing: are these individual particles or 2D averages? If this refers to a 2D average, they should be referred to as 2D averages.

Lines 167-168: The HD, LNK and ILD domains need to be defined, as well as all the ARM domains.

Lines 283: "One of these Ca²⁺-binding sites, named Ca-IVL, is shaped by the side chains of residues D2551, G2550, R2544, H2541, S2545 contributed across the interface between two neighboring subunits (Fig. 3)".

The fact that there is an electron density is not sufficient to qualify as a Ca²⁺ ion. In this case the surrounding residues suggest that this may not be a Ca²⁺ binding site; it could simply be a water molecule. The same applies to Ca-V. The authors should consider other alternatives.

Line 387: "...it is very unlikely that IP3R channel exists in Ca²⁺-free state in vivo."

The authors should qualify better this sentence, otherwise it would imply that the Ca-III site is always occupied, negating its Ca²⁺-sensing role.

Line 394: "All three mutations resulted in significantly decreased amplitude of agonist-induced Ca²⁺ release (Figure 6c, e) when compared with the null cell line"

They do not have lower amplitude than the null cell line; please rephrase.

Reviewer #2:

Remarks to the Author:

This is another "tour de force" study from Serusheva lab. IN the previous publication (Fan at al, 2018) her lab resolves structure of IP3R1 in the presence of Adenophostin A and calcium ions using detergent preparation. In the present report structure of IP3R1 incorporated into lipid nanodisk is solved by CryoEM in the presence of physiological activators (IP3, Ca and ATP, CIA structure) and in he presence of saturating Ca concentrations (20 µM), that is in deactivated state. Obtained results provide atomic-level information about changes in IP3R1 conformation during channel opening (Fig 7). Some of the conclusions are validated in functioanl experiments with IP3R1 gating mutants (Fig 6).

This paper is a significant contribution to our mechanistic understanding of IP3R function and will have a major impact on the field.

Reviewer #3:

Remarks to the Author:

The authors present new structural results on IP3R1 analyzed by Cryo-EM. This manuscript details the structural observations of the IP3R1, reconstituted in nanodiscs, under a set of three conditions 1) Nominally calcium-free, 2) Activating concentration of calcium (20 μM), and 3) Activating conditions that include 2 μM calcium plus IP3 and ATP. This study extends previous works in a few ways. First, higher resolution of the focused-refined regions of the ligands-stabilized protein has allowed assignment of densities, which gives a clearer view of the architecture of the cytosolic regulatory domains. Second, the multiple combinations of ligands have allowed a more comprehensive proposal for the complex gating mechanism involved in IP3, Ca, and ATP-dependent and Ca - dependent priming and gating. Third, high-resolution images of putative Ca-binding sites, the ATP binding site, and the IP3 binding site are presented. Overall, these maps provide substantial structural and conceptual advances in our understanding of this complex channel protein that is central to calcium signaling in a wide range of cell types. I have only minor concerns and suggestions as detailed below.

1- The modeling and interpretation of the CIA-IP3R1 pore is quite confusing. The suggestion that the pore may conduct dehydrated calcium is quite provocative. In order for the Ca^{2+} ion to be "peeled" off from water molecules one would expect a series of charged residues aligned along the selectivity filter (similar to voltage-gated K^{+} channels), not the case in calcium release channels. This should be carefully worded and perhaps an alternative explanation should be explored.

2- In the case of RyR1, the pore aperture is dilated via the bending of S6 outward from the 4-fold symmetry axis and not the turning of it. One would expect the turning of S6 would turn I2590 along with F2586. Again, quite confusing and requires careful observation of the maps and perhaps difference-maps between the CIA and Ca-IP3R1 will reveal a clearer answer. This is really hard to review without actual access to the maps and models.

3- The coordination of the 3 phosphates of the IP3 molecule coordination seems to be confusing too, for example, P5 coordinated/stabilized by 6 residues, was this validated somehow? Again, very hard to review without looking at the maps and models.

4- "...strong non-protein densities" interpreted as Ca^{2+} requires extensive validation. Stronger densities could be many things. Difference maps between the three states could help here too.

Response to Reviewers

We thank reviewers for their overall positive comments and insightful suggestions. We have modified our manuscript (yellow highlight) in response to their suggestions. Our point-by-point responses are provided below.

Reviewer #1:

In this manuscript Fan et al. determined the near-atomic structure (3.5 and 3.26 Å) of IP3R1 in the presence of activating Ca+IP3+ATP and in the presence of inhibiting Ca. They localized the ATP, IP3, and Ca binding sites, with good agreement with existing near-atomic 3D structures of the IP3R-3 isoform (by the Hite and Karakas groups). Additionally, they identified new putative Ca binding sites and for the IP3R-Ca dataset, they also potentially found an inhibitory site for Ca. Some of the sites were validated using functional experiments. Variability analysis identified two conformations of the ARM2 domain, “extended” and “retracted”, where an effect of IP3 is to increase the amplitude of the movement of the ARM2 domain.

MAJOR ISSUES

A major problem is the disconnect between the title/abstract and the rest of the manuscript. The application of the deep neural network method, developed by the team, is not obvious in the manuscript, and the dynamic aspect is quite scant. As such, the manuscript focuses mainly on the different binding sites and on the structure of the pore.

Authors' reply:

We have changed the title and abstract to moderate the focus on the deep-learning method, though it was an important aspect of the study. The manuscript, appropriately we feel, focuses primarily on the biological conclusions of the study and not the technical aspects of how it was achieved. The deep-learning methodology was not developed as part of this manuscript, only used to achieve the results. It primarily focused on understanding the dynamics of the ARM2 domain and critical interactions with the ligand-binding domains (lines 98-102). These results are shown clearly in supplementary movies, were confirmed using traditional classification methods, and played an important role in the structural/functional relationship described in the manuscript.

MINOR ISSUES

1Q1: Line 24: please spell out GMM

1A1: We have made sure to spell out the abbreviation for Gaussian Mixed Model within the text.

1Q2: Line 154: “each individual 2-D particle “The term is confusing: are these individual particles or 2D averages? If this refers to a 2D average, they should be referred to as 2D averages.

1A2: We aren't clear on where the reviewer's idea that we were working with 2D class averages came from, as this term was not used anywhere in this manuscript and neither is it invoked in the GMM manuscript. Each individual 2D particle is mapped to a 3D Gaussian configuration via the GMM. 3D information is encoded in the network using the ensemble of all 2D particles and their orientations. A subspace with limited dimensionality (as described in the GMM paper, ref. #22) permits mapping individual particles through the subspace to a specific 3D configuration, with some uncertainty depending on noise level. We have adjusted the way we phrase this in the manuscript (lines 157-166) to clarify the approach, but for detailed explanations of the mathematical process, please see reference #22.

1Q3: Lines 167-168: The HD, LNK and ILD domains need to be defined, as well as all the ARM domains.

1A3: We have defined the domain abbreviations within the manuscript.

1Q4: Lines 283: “One of these Ca²⁺-binding sites, named Ca-IV_L, is shaped by the side chains of residues D2551, G2550, R2544, H2541, S2545 contributed across the interface between two neighboring subunits (Fig. 3)”. The fact that there is an electron density is not sufficient to qualify as a Ca²⁺ ion. In this case the surrounding residues suggest that this may not be a Ca²⁺ binding site; it could simply be a water molecule. The same applies to Ca-V. The authors should consider other alternatives.

1A4: We agree with the reviewer that the presence of density is not the only criteria that was used in determining ligand binding and we appreciate the opportunity to clarify this point. We have now elaborated on our criteria for analysis of non-protein densities here and within the manuscript. Briefly, putative positions of bound Ca²⁺ ions were initially identified computationally using Phenix and pw_ligands.py (see Methods). Placement of Ca²⁺ ions within an cryoEM density map were based on 1) presence and statistical significance of a density under appropriate experimental conditions where free Ca²⁺ would be available for binding; 2) the absence of the same density under conditions where the ion would be sequestered/chelated rendering it unavailable to bind the specific protein site; 3) reasonably positioned atomic charges within the protein model that could be available/accessible to coordinate the atom/ion and 4) previously predicted putative Ca²⁺ binding sites and physiologically relevant functional studies that support the validation of ascribing the Ca²⁺ ion to the corresponding density. Please, also see below our answer to 3Q4.

1Q5: Line 387: “...it is very unlikely that IP3R channel exists in Ca²⁺-free state in vivo.” The authors should qualify better this sentence, otherwise it would imply that the Ca-III site is always occupied, negating its Ca²⁺-sensing role.

1A5: We thank the reviewer for this good point and clarified this sentence in the revised manuscript (lines 402-406).

1A6: Line 394: "All three mutations resulted in significantly decreased amplitude of agonist-induced Ca²⁺ release (Figure 6c, e) when compared with the null cell line"

They do not have lower amplitude than the null cell line; please rephrase.

1A6: We have revised this statement within the manuscript (lines 413-415).

Reviewer #2 (Remarks to the Author):

This is another "tour de force" study from Serysheva lab. In the previous publication (Fan et al, 2018) her lab resolves the structure of IP3R1 in the presence of Adenophostin A and calcium ions using detergent preparation. In the present report structure of IP3R1 incorporated into lipid nanodisk is solved by CryoEM in the presence of physiological activators (IP3, Ca and ATP, CIA structure) and in the presence of saturating Ca concentrations (20 μ M), that is in deactivated state. Obtained results provide atomic-level information about changes in IP3R1 conformation during channel opening (Fig 7). Some of the conclusions are validated in functional experiments with IP3R1 gating mutants (Fig 6). This paper is a significant contribution to our mechanistic understanding of IP3R function and will have a major impact on the field.

1A: Thank you! We appreciate the reviewer's overwhelmingly positive comments on our manuscript.

Reviewer #3 (Remarks to the Author):

The authors present new structural results on IP3R1 analyzed by Cryo-EM. This manuscript details the structural observations of the IP3R1, reconstituted in nanodiscs, under a set of three conditions 1) Nominally calcium-free, 2) Activating concentration of calcium (20 μ M), and 3) Activating conditions that include 2 μ M calcium plus IP3 and ATP. This study extends previous works in a few ways. **First**, higher resolution of the focused-refined regions of the ligands-stabilized protein has allowed assignment of densities, which gives a clearer view of the architecture of the cytosolic regulatory domains. **Second**, the multiple combinations of ligands have allowed a more comprehensive proposal for the complex gating mechanism involved in IP3, Ca, and ATP-dependent and Ca -dependent priming and gating. **Third**, high-resolution images of putative Ca-binding sites, the ATP binding site, and the IP3 binding site are presented. Overall, these maps provide **substantial structural and conceptual advances in our understanding of this complex** channel protein that is central to calcium signaling in a wide range of cell types.

I have only **minor concerns** and suggestions as detailed below.

We thank this reviewer for overall positive and helpful comments.

3Q1: The modeling and interpretation of the CIA-IP3R1 pore is quite confusing. The suggestion that **the pore may conduct dehydrated calcium is quite provocative**. In order for the Ca^{2+} ion to be “peeled” off from water molecules one would expect a series of charged residues aligned along the selectivity filter (similar to voltage-gated K^+ channels), not the case in calcium release channels. This should be carefully worded and perhaps an alternative explanation should be explored.

3A1: We appreciate the reviewer for pointing this out and we agree that our statement that the hydrophobic ring at I2590 “..is too narrow at this location for passage of hydrated Ca^{2+} , but *it is wide enough for dehydrated or partially hydrated Ca^{2+} ions.*” is misleading. We have clarified this point in the revised manuscript (lines: 369-373). In addition, in the Discussion section we had stated that “...the observed degree of pore opening at I2590 in the CIA structure is not as wide as a hydrated Ca^{2+} ion, it might provide a Ca^{2+} ion passage using a different mechanism” relying on dynamic side chains. Moreover, the pore in structures of several K^+ channels and the human CLC-1 chloride channel are also not wider than a hydrated ion under activating conditions (see references #50-53 in the manuscript). Thus, it is conceivable that a slightly narrow tunnel lined with dynamic side chains can provide a good passage for hydrated ions. It seems that the original assumption that the channel gate must open (‘iris diaphragm’ mechanism) as wide as a hydrated ion, may not be the full story.

3Q2: In the case of RyR1, the pore aperture is dilated via the bending of S6 outward from the 4-fold symmetry axis and not the turning of it. One would expect the turning of S6 would turn I2590 along with F2586. Again, quite confusing and requires careful observation of the maps and perhaps difference-maps between the CIA and Ca-IP3R1 will reveal a clearer answer. This is really hard to review without actual access to the maps and models.

3A2: As we proposed in the manuscript (lines 372-373; 464-468), it is conceivable that dynamic side chains surrounding the ion conduction path play a role in regulating the passage of hydrated ions through the ion conduction pore. We also would like to point out that despite of structure-function conservations described within tetrameric Ca^{2+} channels, the precise gating mechanism might not be conserved between IP_3R and RyR channels. In fact, similar diversity in gating mechanisms is observed among tetrameric TRP channels. Moreover, the ligand-bound RyR1 structures solved in detergent-based water environment might exhibit gating motions that are different from that in ligand-bound $\text{IP}_3\text{R1}$ reconstituted in lipid nanodiscs solved in the present study.

3Q3: The coordination of the 3 phosphates of the IP_3 molecule coordination seems to be confusing too, for example, P5 coordinated/stabilized by 6 residues, was this validated somehow? Again, very hard to review without looking at the maps and models.

3A3: We thank the reviewer for pointing this out. The molecular interactions of IP₃ within the IP₃R1 binding pocket are supported by previous X-ray crystallography of expressed IP₃ ligand-binding domains and extensive mutagenesis studies (as cited in the manuscript references #16-18,29-32,35) and are in-line with the determinants presented in our current and previous (Fan et al., 2018, <https://doi.org/10.1038/s41422-018-0108-5>) structural studies of the tetrameric, full-length IP₃R1 channel. We have clarified the description of IP₃ binding pocket determinants (lines 195-200).

3Q4: “...strong non-protein densities” interpreted as Ca²⁺ requires extensive validation. Stronger densities could be many things. Difference maps between the three states **could** help here too.

3A4: Yes, we agree with the reviewer that non-protein densities can be difficult to accurately define. Detailed description for the methods and criteria used to identify protein-bound ligand is now provided in Methods; please also see our reply to 1Q4. Briefly, we use an initial computational step to identify potential ligand density; a difference map is created based on the original density map and model. Non-modeled density is then checked for certain distance, chemical and geometrical constraints. Similar approaches are used in other Phenix programs (e.g., Dang et al., 2017, <https://doi.org/10.1038/nature25024>); pw_ligands.py is built upon our expertise with our modeling software, Pathwalking, which is also in Phenix. From this computational assignment, we then evaluated all of the putative positions manually, paying particular attention to chemistry and previously suggested binding sites.

Reviewers' Comments:

Reviewer #1:

Remarks to the Author:

The answers provided by the authors, together with the changes introduced in the revised manuscript, addressed all my concerns. The structural work together with the conformational analysis, provide new concepts that help to understand the complex regulation by multiple ligands. Overall, this work represents an important advancement in the field.

Reviewer #3:

Remarks to the Author:

The authors have addressed well my concerns.

I recommend publishing it in Nature Communications without further changes.